# Evaluation of Ecological Environment Effect of Villages Land Use and Cover Change: A Case Study of Some Villages in Yudian Town, Guangshui City, Hubei Province

**Wei Ren [1,2], Xuesong Zhang [1,2,*] and Yebo Shi [1,2]**

[1] Hubei Province Key Laboratory for Geographical Process Analysis and Simulation, Wuhan 430079, China; renwei@mail.ccnu.edu.cn (W.R.); shiyebo@mails.ccnu.edu.cn (Y.S.)

[2] College of Urban and Environmental Sciences, Central China Normal University, Wuhan 430079, China

\* Correspondence: zhangxuesong@mail.ccnu.edu.cn

**Abstract:** Rapid economic development has a significant negative impact on the rural ecological environment. Evaluating the ecological environmental effect of land use and its change trend at the village scale has important practical significance for maintaining ecological functions and ensuring ecological safety. Taking a typical village in Yudian Town as an example, we applied a land-use ecological environment effect evaluation and the CA-Markov change trend prediction model and constructed an index of ecological environmental effect status. Based on the land use, resource environment, and social economic data from 2014 and 2019, we evaluated the ecological environmental effects of land use in each village, simulated the land-use change in each village in two different scenarios, i.e., the developmental orientation (DO) and ecological orientation (EO), in 2030, and analyzed the corresponding change trend of the land-use effect. The ecological environmental effect of land use showed obvious characteristic differentiation in villages with different development levels. For example, villages with poor natural geographic background conditions and slower economic development had a good level of ecological environmental effect, whereas villages with better resource and environmental endowments but faster economic development had lower levels of ecological environmental effect. Village land-use management methods have had a certain effect on improving ecological security, but the effect has been slow. In conclusion, the research results portray the relationship between rural land use and ecological environmental effects in low hilly areas in northern Hubei at a small scale and have reference value for land resource allocation and spatial pattern optimization in similar regions.

**Keywords:** ecological environment effect evaluation; land use; CA-Markov model prediction; villages

## 1. Introduction

Land use and cover change (LUCC) is the material basis, energy source, space carrier, and constituent element of ecological civilization construction, and supports the development of all walks in life, having an overall, strategic and fundamental position in the modernization drive. Since the 1990s, resource, environment, and population issues have become increasingly prominent, and land use and cover change research has become the frontier of global change research [1,2]. Land use refers to the purposeful development and utilization of land resources by humans, and land cover refers to the natural or man-made coverage of the surface. LUCC not only objectively records the spatial pattern of humans changing the characteristics of the Earth's surface but also reproduces the temporal and spatial dynamics of the Earth's surface landscape. Due to the deepening of research on global environmental change, various countries have gradually realized that it is an important cause of global environmental changes. LUCC is closely related to the ecological environment [3–7]. After entering the 21st century, with the direction of ecological environmental change research and the rapid development of rural areas, land use and cover change in

villages has become the core theme of current global rural ecological environmental change research [8].

LUCC at the regional scale causes changes in the local ecological environment, which is an important component of ecological environmental changes in the region [9,10]. Riebsame et al. [11] believed that natural and social factors are the main causes of land-use change. Fu et al. [12] noted that climate change and human activities are the main factors affecting land-use change. Bicík et al. [13] believed that social economy, policy, technology, natural factors, and cultural factors are the main factors affecting land use. Therefore, land use is not only restricted by natural factors but also influenced by social, economic, technological, and historical factors, and it is also comprehensive and regional. More scholars have extensively studied the effects of regional land use and cover change and regional climate, hydrology, soil nutrients, ecosystem services, and biodiversity. Among them, Newbold et al. [14] discussed the interaction between land use and climate change in a terrestrial system. Olson et al. [15] used two empirical models to predict the natural background of static (e.g., geological and soil) and dynamic (e.g., climate and vegetation) environmental factors. The results showed that most land-use changes were related to the increase in human land use but not to regional climate change. Huang et al. [16] used land-use data and the C-Fix model to influence runoff and soil loss, and found that the main basis for affecting change in the ecological environment was soil loss. Thus, it is important to adopt appropriate strategies to control soil erosion. In the research of Tiwari et al. [17], four types of land use/land types (natural forest, mixed forest, prairie, and agricultural land) were used and the soil microbial biomass was studied in land-use changes at different soil depths; it was found that soil microbial biomass is regarded as the key to the soil fertility index, and land use is another main cause of soil microbial community composition/biomass loss in the area. Woldeyohannes et al. [18] assessed the impact of land use and cover change dynamics in southern Ethiopia based on the value of ecosystem services from 1985 to 2050. Enhancing the natural capital of the watershed (e.g., protecting natural vegetation and water bodies) can restore degraded ecosystems.

In recent years, scholars have focused on building an ecological environmental effect index system to analyze the impact of land-use change on the ecological environment, and the research focus has gradually shifted from the comprehensive ecological environmental effect of land use to the ecological environmental response of land-use changes. Liu et al. [19] analyzed the ecological environmental effect mechanism of land consolidation through the logical framework of land consolidation project–land natural elements/land use-types–land landscape elements–land ecological service function. Guo et al. [20] analyzed the increases and decreases in various types of land and the degree of dynamic change in the loess hilly region of western Shanxi and used indicators such as the land cover status index, ecological environmental quality index, and ecological contribution rate of land transformation to analyze the ecological environment effect of land-use change. Dai et al. [21] analyzed land-use change and resource ecological effects on three typical mountainous areas in China: Taihang Mountain, Hengduan Mountain, and Guizhou-Guangxi Karst Mountain. The results showed that the spatial heterogeneity of mountainous land and its corresponding resource ecological effect were key to the sustainable development of mountainous areas. Hu et al. [22] analyzed the spatial pattern characteristics of the economic, social, ecological, and comprehensive benefits of land use in Jiangsu Province by constructing a land-use benefit evaluation index system using the natural break point method and the hot spot analysis method. Shen et al. [23] analyzed the landscape ecological risk assessment model under land use and cover change through the three phases of Landsat remote sensing images in 2001, 2010, and 2019, and studied the land-use change, landscape ecological risk assessment, and its temporal and spatial differentiation in the lower reaches of the Tarim River.

Overall, the current evaluation of the ecological environmental effect of land-use change has focused more on the influence of single environmental factors [24–27]; however, social and economic pressures have been less considered, the research scales have been

mainly concentrated in regions and places, and the research has been mostly based on a single land-use type change, area change, center of gravity change, ecological environmental quality index, and ecological risk [28–30]. Few people combine these factors to analyze and predict the ecological environmental effect of land-use change on a small scale. Hubei Province is one of the main production areas in China and is the core area for the development of the Yangtze River Economic Belt. A comprehensive evaluation of land-use change and ecological effects is of great practical significance for ensuring food security and promoting the construction of ecological civilization in the Yangtze River Economic Belt. Thus, this research takes the typical village of Yudian town, Guangshui city, which is located in the low hills of northern Hubei Province, as the study area, selects the two phases of remote sensing image data in 2014 and 2019, and uses geographic information software such as ArcGIS and ENVI for this area. On this basis, we simulated the land-use change in 2030, predicted the corresponding ecological environmental effect change trend, analyzed the impact mechanism of land-use change on the ecological environment, and then provide references and suggestions for regional land and space planning and land consolidation practices.

## 2. Concept and Evaluation Ideas

### 2.1. Concept Analysis of Ecological Environmental Effect

The concept of ecological effects originates with the cognition of ecological security and ecological risk. Ecological security is a complex conceptual issue that can be divided into broad and narrow senses. Broadly speaking, it refers to the changes in the structure and function of the ecosystem caused by environmental pollution and environmental damage caused by human activities. The security status of a composite ecosystem can effectively protect human life and health from damage to ecological environmental conditions and ecosystem services, such that economic development and social stability are not hindered or threatened. In the narrow sense, starting from nature itself, it refers to the integrity and health of its own structure required by the ecosystem to maintain biodiversity and perform ecosystem functions [31]. The concept of ecological risk applies to the ecosystem and its components. It assesses and predicts the threat of human activities or adverse events to the ecological environment and the impact of the possibility of adverse factors, in addition to the evaluation of the technical method system used to determine the acceptable degree of ecological risk. In contrast, ecological risk research is more purposeful and accurate. It is often carried out for a specific group under a certain condition and has widely accepted and specific research methods. Ecological security research is more focused on changes at the regional scale, changes in ecological patterns, changes in ecological processes, and their overall effect. The wide range and complexity of research objects makes ecological security research more focused on conclusions rather than absolutely qualitative. The commonality between ecological security and ecological risk lies in the enhancement of the stability of ecological patterns and ecological processes [32].

The ecological environment refers to the general term for the quantity and quality of water resources, land resources, biological resources, and climate resources that affect the survival and development of human beings. It is a complex ecosystem related to the sustainable development of society and the economy. Ecological environmental problems refer to the various negative feedback effects that harm human survival; these effects are caused by the destruction and pollution of the natural environment in the process of using and transforming nature for survival and development [33]. Ecological environmental effects refer to the changes in the ecological processes of various components, components in the ecosystem under certain pressure, and the resulting changes in the regional ecological pattern. The impact of this change may be positive (ecological security) or negative (ecological risk). Therefore, the ecological environmental effect can be regarded as a more comprehensive overall assessment of the stability of the ecosystem. As a seminatural landscape area dominated by man-made activities, the village encompasses a three-life space (production, life, and ecology) and has a profound impact on the method and intensity of

land use. This kind of influence continues to accumulate and is directly reflected in the structure and composition of the ecosystem, which is reflected in the ecological environmental effect of land use. In the process of village development, disordered construction activities, lack of control over waste discharge, and destruction of natural vegetation may all cause ecological risks. Therefore, planning and arrangement, construction management, and other means are often used to reduce the ecological risks caused by human activities and increase the positive impact on ecological security (Figure 1).

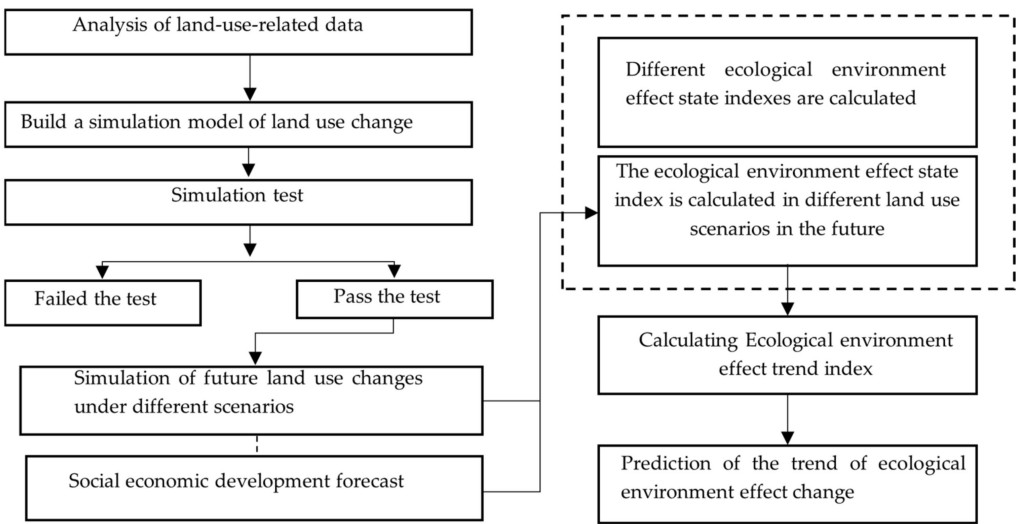

**Figure 1.** Land-use ecological environmental effect model.

*2.2. Ecological Environmental Effect Evaluation Characteristics*

The research scale of ecological environmental effect evaluation is regional and typical, and the research questions are systematic and regular. The ecological environmental effect considers humans as the dominant unit, starting from the perspectives of resources, environment, and humanities, and analyzes the initiative a priori of the ecosystem under the natural state [34]. Humans should lack absolute unified standards, pay more attention to sustainability and safety, and emphasize the ability to maintain the stability of the ecosystem within a certain period of time [35]. In terms of research content, the ecological environmental effect evaluation is consistent with the research on ecological security and ecological risk. This is divided into two different dimensions based on patterns and processes that correspond to the changes in the static distribution and dynamic evolution process of each component in the ecosystem caused by external pressure [36].

*2.3. Evaluation Ideas*

2.3.1. Index System Construction

The assessment of the ecological environment should be composed of resources, the environment, and social and economic indicators related to land-use change. Resources emphasize that ecosystems provide humans with a long-term stable and affluent natural capital base; environment emphasizes that ecosystems provide humans with environmental elements such as water, soil, and organisms; society and economy mainly reflect the change in and development of local land use and whether this change is conducive to ensuring a reasonable layout of ecological land. The evaluation of ecological environmental effects should focus on the stability and sustainability of the ecosystem. This paper mainly discusses four components that have direct impacts on the ecological environmental effect. The selection of indicators is also based on the ecological environment.

2.3.2. Technical Route of Ecological Environmental Effect Evaluation

The ecological environmental effect includes two dimensions: pattern and process. It is both a state quantity and a process quantity. To measure the two, the ecological

environmental effect status index and the ecological environmental effect trend index were constructed. Of these, the ecological environmental effect state index represents the current state of each ecosystem component related to the village's land use at a certain point in time, and is composed of the resource effect state index, the environmental effect state index, the economic effect state index, and the social effect state index. The ecological environmental effect trend index represents the change in the ecological pattern caused by the change in the village's land use over a period of time. It is obtained by calculating the difference in the ecological environmental effect status index in different years and is correspondingly decomposed into four trend indexes. To make the evaluation results conform to conventional expression habits and easy to understand, according to the calculation results of the state index and trend index, the ecological environment effect status types are classified into four categories: good, stable, declining, and degraded, and the ecological environmental effect trend types are classified as significantly positive. There are five categories: significantly positive, slightly positive, basically stable, slightly negative, and significantly negative. (Figure 2).

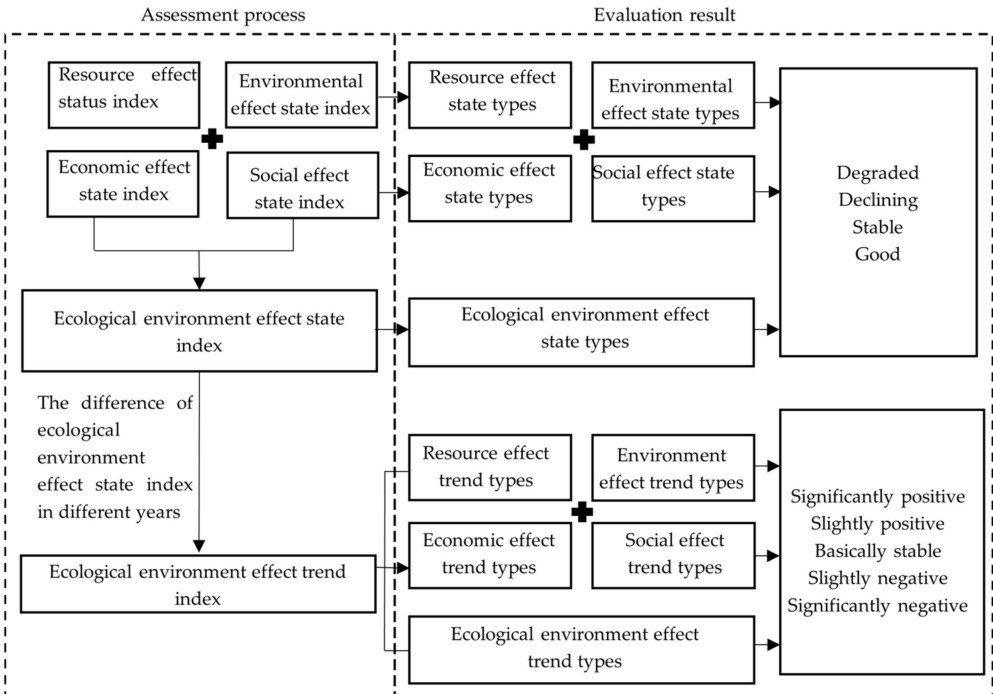

**Figure 2.** Overall thoughts on the evaluation of ecological environmental effect of land use.

### 2.3.3. Ecological Environmental Effect Trend Forecast

The objective of an ecological environmental effect evaluation is to determine both the current status of land use and its evolution trend. Therefore, it is necessary to predict the development trend of future changes in land use [33]. Combining the ecological environmental effect with land management, land-use planning is used as a basis, differentiated development prospects are established, and land-use change is simulated. Based on simulating the ecological environmental effect of land-use change in different situations, the evolution trend of the ecological environment effect of land use in the process of village development can be realized. (Figure 3).

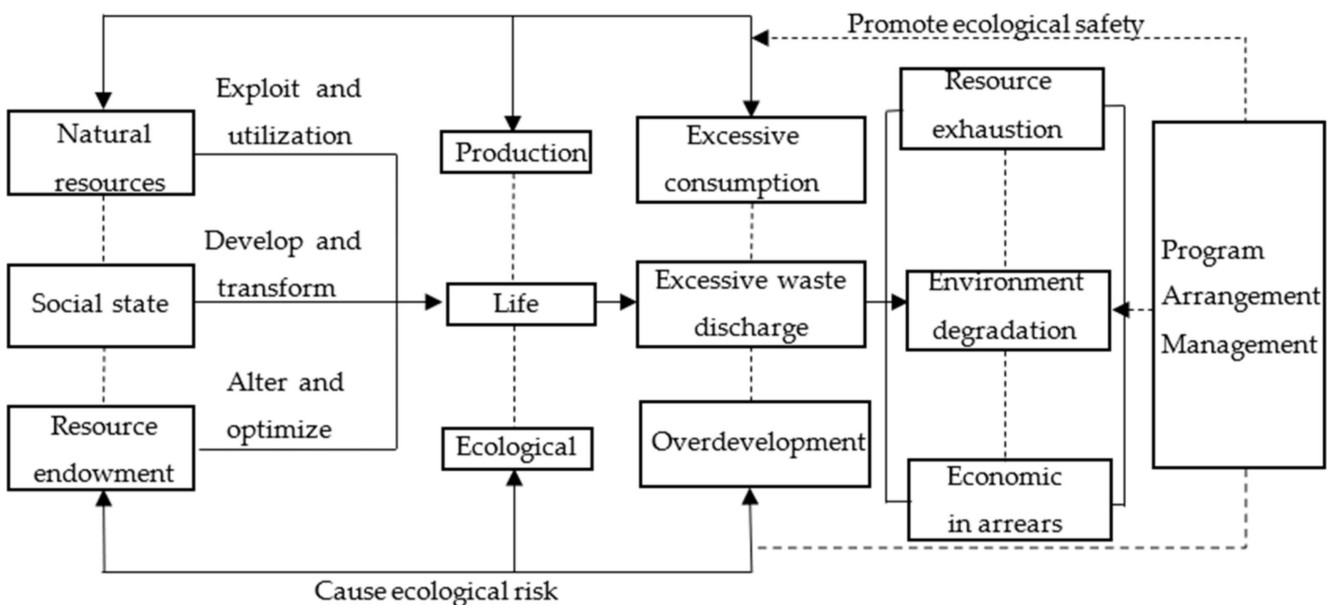

**Figure 3.** Prediction of the change trend of the ecological environmental effect of land use.

## 3. Data and Methods

### 3.1. Overview of Study Area

Guangshui city is located in the eastern part of the northern part of Hubei Province; it is part of Suizhou city and is a county-level city. Yudian town is located in the northwestern part of Guangshui city on the east bank of the Xianjue Temple Reservoir; in the west, it faces the Zengdu District of Suizhou city. It has a north subtropical continental monsoon climate. The annual average temperature has ranges of 13–16 °C and 13–14 °C in the northern region, 14–15 °C in the middle area, and 15–16 °C in the southern area, and the temperature difference between the north and south is 2 °C. The average frost-free period is between 201 and 240 days, and the average annual rainfall is between 940 and 1040 mm. It extends across 113°31′–114°07′ E and 31°23′–32°05′ N [37]. The study area includes eight administrative villages: Lufan Village, Gucheng Village, Wangjiachong Village, Jinpan Village, Shuanglou Village, Yingzizhai Village, Baique Village, and Baoziling Village. There are 2955 households in the eight villages and 99 natural bays, including 348 households in Lufan Village, 298 households in Gucheng Village, 334 households in Wangjiachong Village, 421 households in Jinpan Village, 478 households in Shuanglou Village, 368 households in Yingzizhai Village, 386 households in Baique village, and 322 households in Baoziling Village. The total registered population in the study area is 12,325, the permanent population is 3,206, and the per capita disposable income is 11,000 yuan, which mainly comes from migrant workers. The study area has high-quality agricultural products, such as white eggplant, white cucumber, and Chinese cabbage, in addition to national-level protected tree species such as camphor, Nanmu, ginkgo, and cephalotaxus; the medicinal materials include centipede, wormwood, platycodon, and *Prunella vulgaris*; and protected wild animals include red fox, flower-faced raccoon, skylark, and white crane. The regional characteristics of the study area are as follows. The natural resources and economic status are universal in the low hilly and hilly rural areas of northern Hubei, and the region's development has typical exemplary significance for the construction of ecological civilization and rural revitalization in similar regions (Figure 4).

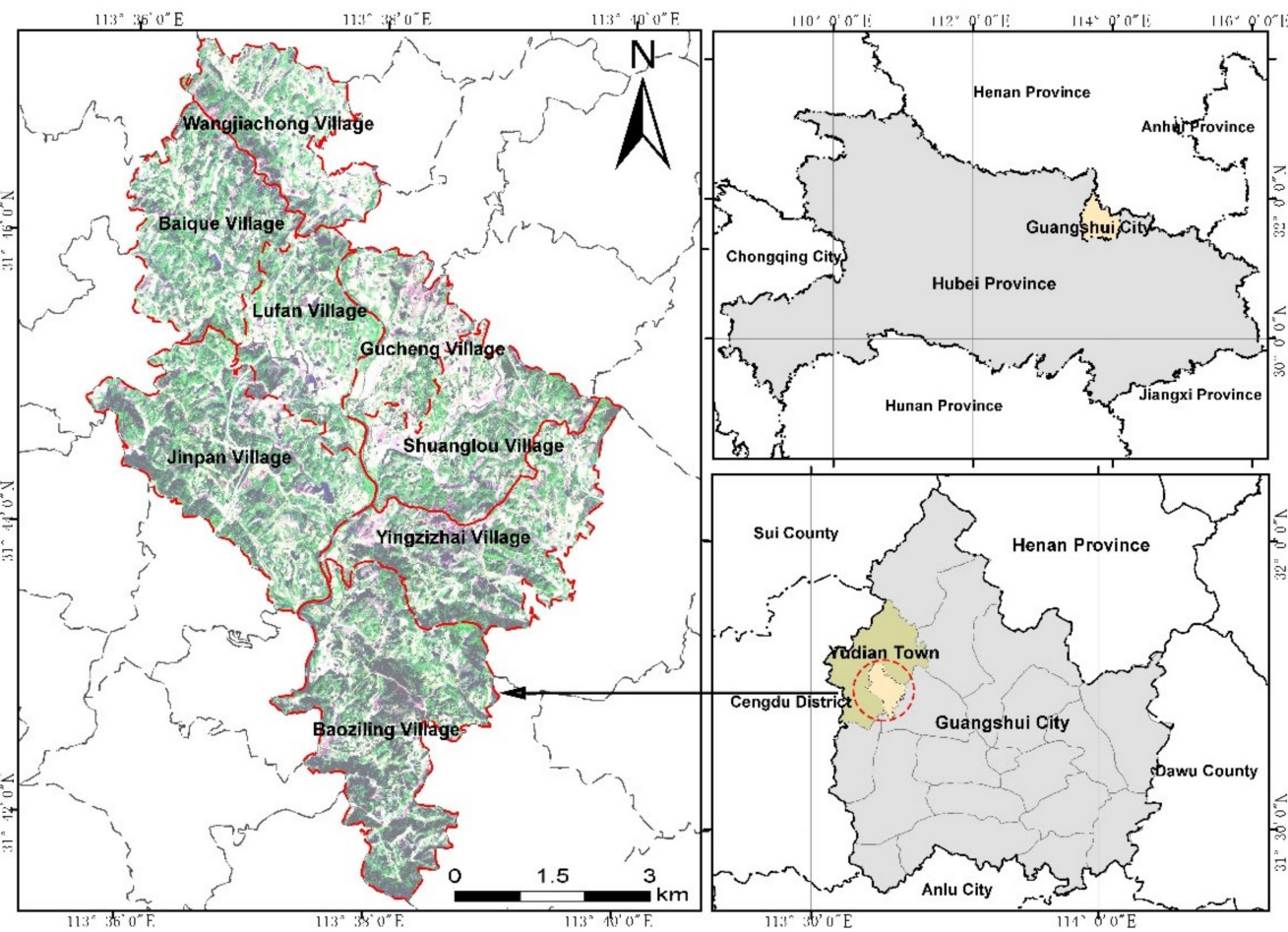

**Figure 4.** Location map of the study area (eight administrative villages).

### 3.2. Land-Use Status

According to the survey data of changes in land-use status in the study area in 2019, the total land-use area in this area is 3373.79 hectares. There were 1231.41 hectares of arable land, accounting for 36.49% of the total land area, of which 878.98 hectares were paddy fields and 326.34 hectares were dry land. The arable land in the area accounts for a large proportion of the land-use structure and has good arable land planting resources. The garden area is 13.95 hectares, accounting for 0.41% of the total land area; the forest land is 1247.41 hectares, accounting for 36.97% of the total land area; and the grassland amounts to 409.93 hectares, accounting for 12.14% of the total land area. The areas of forest and grass are relatively large, reflecting the fact that the region has rich ecological resources and is suitable for the development of modern agriculture and tourism. The residential land is 190.96 hectares, accounting for 5.66% of the total land area, and the land-use efficiency is not high. Public management and public service land, industrial and mining storage land, transportation land, water area and water conservancy facility land, other land, and special land account for 0.08%, 0.09%, 1.55%, 6.36%, 0.22%, and 0.02% of the total land area, respectively (Figure 5).

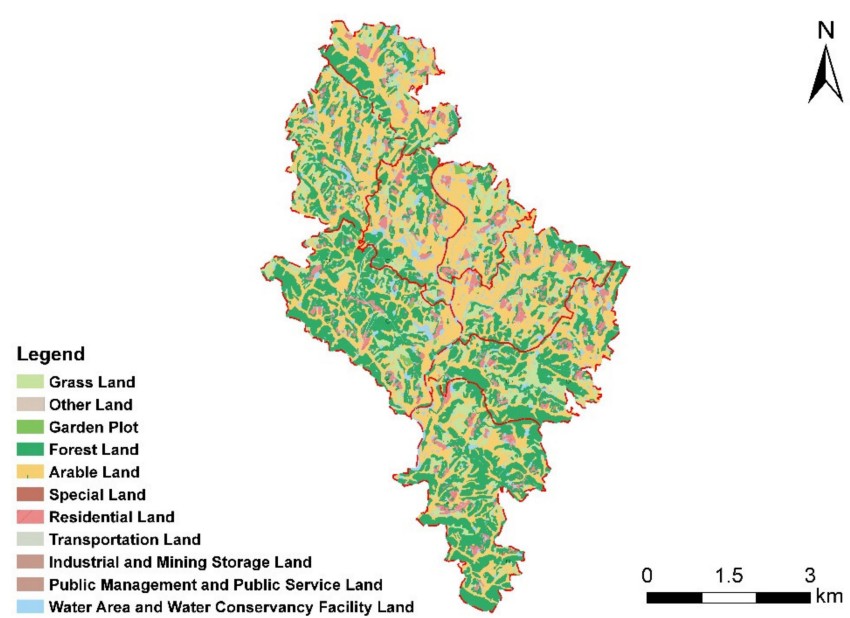

**Figure 5.** Overview map of land-use status.

*3.3. Evaluation of the Ecological Environmental Effect Based on the Status of Land Use*

3.3.1. Ecological Environmental Effect Evaluation Index

We read relevant periodicals and literature and combined the monographs in related fields, national standards, and industry standards to identify the commonly used indicators related to key characterizations of land-use ecological environmental effects and form a high-frequency indicator database through frequency statistics [38–40]. According to the actual situation of the study area, the following three criteria were used to screen high-frequency indicators: (1) difference: the indicators should reflect certain differences at the village scale; (2) feasibility: the basic data involved in the indicators have better accessibility; and (3) continuity: the available indicators require data continuity to meet the needs of longitudinal comparison of evaluation results. The resource effect status index was constructed with four indicators: the intensity of national land development, per capita construction land area, per capita arable land area, and per capita forest area. The environment effect status index was constructed with four indicators: water network density, vegetation coverage, land degradation, and biodiversity. The social effect status index was constructed with three indicators: population density, population natural growth rate, and the Engel coefficient. The economic effect status index was constructed with three indicators: land for transportation, per capita public budget input, and per capita GDP value. Finally, the ecological environmental effect evaluation index system of land use in the region was determined (Table 1).

3.3.2. Data Collection and Processing

Due to the completeness of the statistical data in the study area, the time point for the evaluation of the ecological environmental effect of the land use status was set to 2019, and the period of the evaluation of the trend of the ecological environmental effect was set to 2014–2019. The basic data for the evaluation were taken from (1) the 2014 and 2019 land-use change survey database (1:10,000) provided by the Natural Resources and Planning Bureau where the study area is located; (2) the basic physical geography data of the study area (including topography (e.g., basic data such as hydrology and vegetation); (3) the Statistical Yearbook of the location of the study area (2014–2019); (4) 314 valid questionnaires of the socioeconomic survey based on the study area (2019); (5) the 2014 and land-use data obtained by human computer interaction interpretation of the Gaofen-2 (GF-2) satellite remote sensing image (spatial resolution 4 m) in 2019.

**Table 1.** Evaluation index system of the ecological environmental effect of land use.

| Total Index | Each Index | Index | Index Weight | Index Attributes |
|---|---|---|---|---|
| Ecological Environment State Index | Resource Effect State Index | Intensity of national land development | 0.0916 | + |
| | | Per capita construction land area | 0.0597 | − |
| | | Per capita arable land area | 0.0717 | + |
| | | Per capita forest area | 0.0683 | + |
| | Environmental Effect State Index | Water network density | 0.0689 | + |
| | | Vegetation coverage | 0.0737 | + |
| | | Land degradation | 0.0484 | − |
| | | Biodiversity | 0.0683 | + |
| | Social Effect State Index | Population density | 0.0596 | + |
| | | Population natural growth rate | 0.0526 | + |
| | | Engel coefficient | 0.0632 | − |
| | Economic Effect State Index | Land for transportation | 0.0627 | + |
| | | Per capita public budget input | 0.0698 | + |
| | | Per capita GDP value | 0.0791 | + |

### 3.3.3. Weight Calculation Method

The normalization method was used to process the initial values of the indicators, the weights of the indicators were assigned with hierarchical weighting, the judgment matrix was constructed, and the analytic hierarchy process (AHP) was used to calculate the weight of each indicator (Table 2):

$$\lambda_{ij} = \frac{100}{\max(v\prime_{ij})}, v_{ij} = \lambda_{ij} \bullet v\prime_{ij} \tag{1}$$

where $\lambda_{ij}$ is the normalized coefficient of index $j$ of index $i$, $V_{ij}$ is the initial value of the corresponding index, and $V_{ij}$ is the normalized value. The initial values of each individual index were calculated and normalized, and the resource effect state index, environmental effect state index, social effect state index, economic effect state index, and ecological and environmental effect state index were calculated after weighting:

$$M_R = \sum_{i=1}^{n} w_i \bullet v_i (i, j = 1, 2, 3 \dots n) \tag{2}$$

$$M_E = \sum_{i=1}^{n} w_i \bullet v_j (i, j = 1, 2, 3 \dots n) \tag{3}$$

$$M_s = \sum_{i=1}^{n} w_i \bullet v_i (i, j = 1, 2, 3 \dots n) \tag{4}$$

$$M_C = \sum_{i=1}^{n} w_i \bullet v_j (i, j = 1, 2, 3 \dots n) \tag{5}$$

$$L = \sum (w_R \bullet M_R, w_E \bullet M_E, w_S \bullet M_s, w_C \bullet M_C) \tag{6}$$

where $M_R$ is the resource effect state index, $W_i$ is the weight value of indicator $i$ of the resource effect state index, and $V_i$ is the normalized value of indicator $i$ of the resource effect state index. $M_E$ is the environmental effect state index, $W_j$ is the weight value of indicator $j$ of the environmental effect state index, and $V_j$ is the normalized value of indicator $j$ of the environmental effect state index. $M_S$ is the social effect state index, $W_i$ is the weight value of indicator $i$ of the social effect state index, and $V_i$ is the normalized value of indicator $i$ of the social effect state index. $M_S$ is the economic effect state index, $W_j$ is the weight value of indicator $j$ of the economic effect state index, $V_j$ is the normalized value of indicator $j$ of the economic effect state index; $L$ is the ecological effect state index, $W_R$ is the resource effect state index weight value, $W_E$ is the environmental effect state index weight value,

*Ms* is the social effect state index weight value, and $M_C$ is the economic effect state index weight value.

The resource effect trend index, environmental effect trend index, social effect trend index, economic effect trend index, and eco-environmental effect trend index were obtained by calculating the difference between the relevant index values of each township in 2019 and 2014, calculated as follows:

$$\triangle M_R = M_{R(t_2)} - M_{R(t_1)} \tag{7}$$

$$\triangle M_E = M_{E(t_2)} - M_{E(t_1)} \tag{8}$$

$$\triangle M_S = M_{S(t_2)} - M_{S(t_1)} \tag{9}$$

$$\triangle M_C = M_{C(t_2)} - M_{C(t_1)} \tag{10}$$

$$\triangle L = L_{t2} - L_{t1} \tag{11}$$

where $\Delta M_R$ is the resource effect trend index, $\Delta M_E$ is the environmental effect trend index, $\Delta M_S$ is the social effect trend index, $\Delta M_C$ is the economic effect trend index, $\Delta L$ is the ecological effect trend index, $t_1$ is the initial year (2014), and $t_2$ is the termination year (2019).

### 3.4. Prediction of Ecological Environmental Effect Based on Land Use and Cover Change Simulation

#### 3.4.1. Scene Setting

According to the actual situation of Yudian Town, two different land-use changes were set for developmental orientation (DO) and ecological orientation (EO), in which developmental orientation (DO) was targeted at the 2020 scale of construction land set in the general land-use plan of Yudian Town (2010–2030). Ecological orientation (EO) strictly protects basic farmland and restricts the transformation of ecological land such as grassland and forest land into construction land and other related government requirements.

#### 3.4.2. Model and Verification

According to the actual situation of the study area, the two different land-use change scenarios of developmental orientation (DO) and ecological orientation (EO) were set, and the CA-Markov model was used to simulate the land-use change in the area. The combination of the quantitative predictive Markov model and the CA model, which can simulate spatial evolution in a complex manner, is commonly used in land-use change simulations [41,42] and can effectively predict the spatial changes in land use occurring in a certain time series. Based on the interpreted data of the high-fraction 2 (GF-2) satellite remote sensing image maps of the study area from 2014 to 2019, the CA-Markov model in IDRISI 15.0 software was used to realize the model construction and simulate the land use of the study area in 2030, and the Kappa coefficient was introduced to test the consistency of the simulation results. The Kappa coefficient of the simulation results was 0.7542, which indicated that the model could predict the land-use changes in the study area accurately (Figure 6).

#### 3.4.3. Land-Use Scenario Simulation and Ecological Environmental Effect Index Calculation

The tested CA-Markov model was selected to simulate the land-use changes in the study area in 2030 under developmental orientation (DO) and ecological orientation (EO), obtain the corresponding land-use data, and calculate the initial and normalized values of each indicator in 2030 based on the population and economic development projections of each village. Based on the forecasted economic development of each village in 2030, we calculated the weight indexes of the initial and normalized values of each index under the EO and DO scenarios, calculated the land use ecological effect state indexes under the two scenarios in 2030, and interpolated with the ecological effect state indexes in 2019 to obtain the land use ecological effect trend indexes of each village from 2019 to 2030.

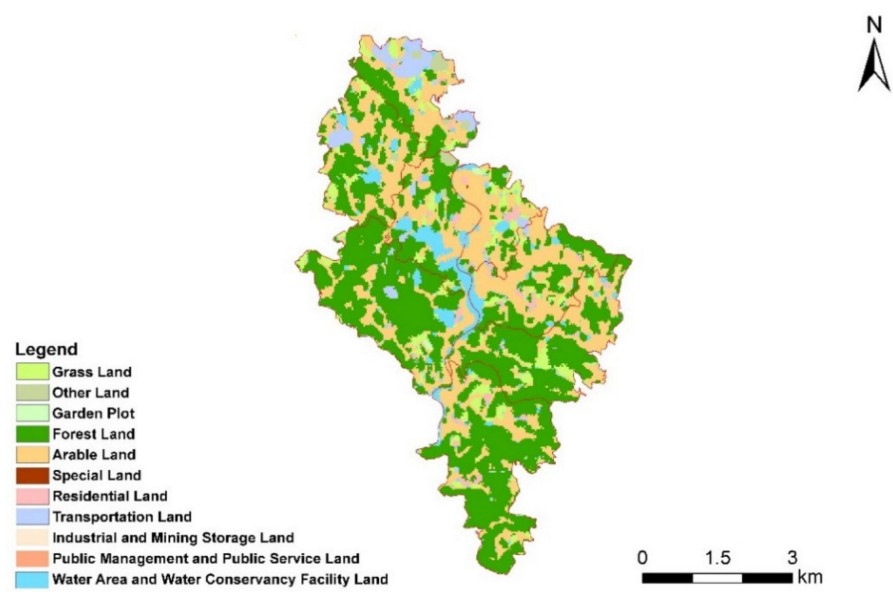

**Figure 6.** Simulation of land use results in 2030.

## 3.5. Evaluation Result Inspection

SPSS 20.0 software was used to calculate the ecological environmental effect status indexes ($M_R$, $M_E$, $M_S$, $M_C$, $L$) and ecological environmental effect trend indexes ($\Delta M_R$, $\Delta M_E$, $\Delta M_S$, $\Delta M_C$, $\Delta L$). Cluster analysis was performed to classify the ecological environmental effect status of each village in the area into five categories: significantly positive, slightly positive, basically stable, gently negative, and significantly negative. The evaluation results are shown in Figures 7–9.

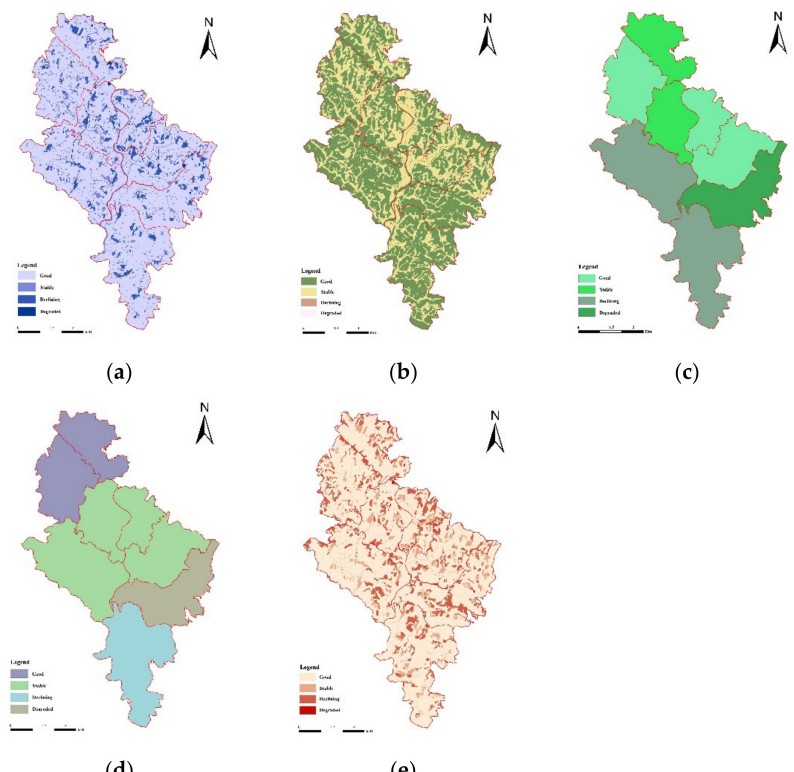

**Figure 7.** Evaluation results of the ecological environmental effect status of rural land use in 2019. (**a**) Resource effect status in 2019; (**b**) environmental effect status in 2019; (**c**) social effect status in 2019; (**d**) economic effect status in 2019; (**e**) ecological environment effect status in 2019.

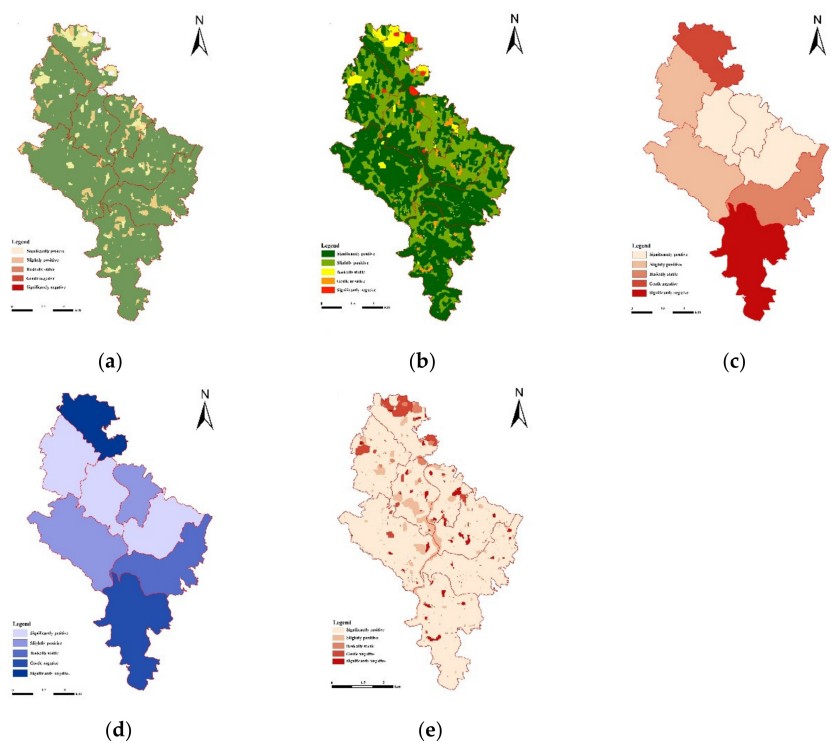

**Figure 8.** Evaluation results of the ecological environmental effect trend of rural land use from 2019 to 2030 (developmental orientation, DO). (**a**) Resource effect trend 2019–2030; (**b**) environmental effect trend 2019–2030; (**c**) social effect trend 2019–2030; (**d**) economic effect trend 2019–2030; (**e**) ecological environmental effect trend 2019–2030.

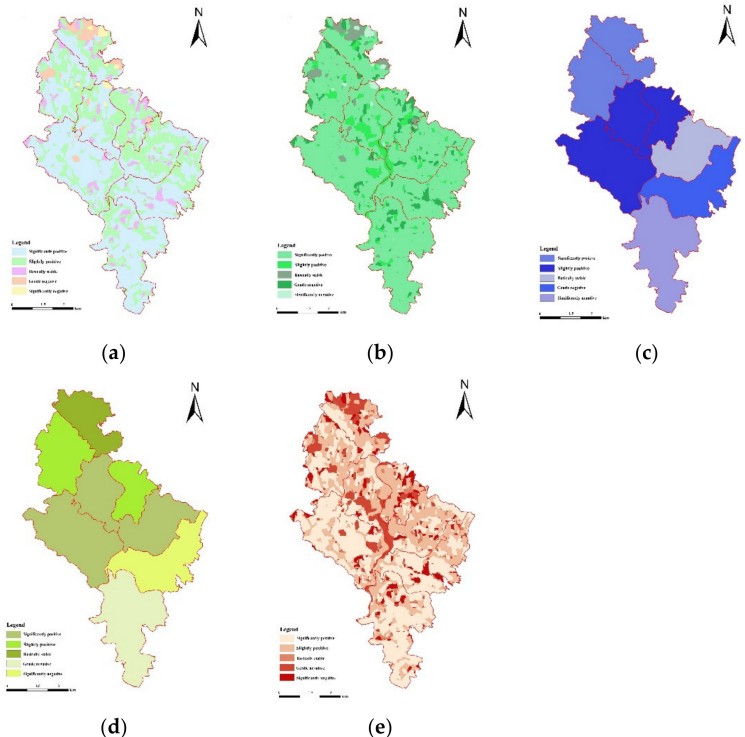

**Figure 9.** Evaluation results of the ecological environmental effect trend of rural land use from 2019 to 2030 (ecological orientation, EO). (**a**) Resource effect trend in 2019–2030; (**b**) environmental effect type in 2019–2030; (**c**) social effect type in 2019–2030; (**d**) economic effect trend in 2019–2030; (**e**) ecological environmental effect trend in 2019–2030.

## 4. Results and Analysis

### *4.1. Analysis of the Ecological Environmental Effect Evaluation Result from 2014 to 2019*

The average value of the ecological environmental effect status index in each village in 2014 was 58.34. In comparison, the higher-altitude villages of Wangjiachong, Baique, and Yingzizhai had better ecological environmental effect status indexes than those of the lower-altitude villages, such as Jinpan, Lufan, Gucheng, Baoziling, and Shuanglou. By 2019, the number of villages with average and poor ecological environmental effect indexes increased significantly, and the level of the original high-altitude villages ecological environmental effect status index declined significantly; additionally, the overall level of the region decreased from 58.34 in 2014 to 56.79 in 2019, showing obvious performance during the period. The ecological environmental effect of the eight villages had a deteriorating trend, and the ecological environmental effect trend type of the five villages was negative evolution; the pattern was as follows: resource effect > environmental effect > social effect > economic effect. By summarizing the evaluation results of the ecological environmental effect of land use in each village from 2014 to 2019, it was found that different development models presented differentiated ecological environmental effects:

1.  Village agglomeration type: Due to the weakening of the village spatial organization and chaotic housing construction, the population, land, industry, and functions were reorganized under the framework of township integration. Some of the eight villages involved in the study area have been merged. After the homestead area of some village groups in Wangjiachong, Baoziling, and Jinpan was concentrated, the original area became large-scale dry land; after the homestead area of some village groups in Baique and Shuanglou was concentrated, the original area became large-scale paddy field and dry land. After the base areas of some village groups in Lufan and Yingzizhai were concentrated, the original area became large-scale dry land and irrigated land. After the area of some village groups in Gucheng was concentrated, the original area became large-scale construction land. After agglomeration, the land utilization rate was greatly increased, thereby alleviating the negative impact of the economic effect status index caused by land problems.

2.  Slow decay type: The northern mountainous areas with poor natural geographical conditions showed the difference between the resource effect state index and the environmental effect state index. For example, Wangjiachong Village and Baique Village had higher resource and environmental effect status indexes than Baoziling Village and Lufan Village. Due to the continuous implementation of tree planting and grass planting projects since the implementation of the "Green Man Jing Chu" (special greening project) project in 2014, vegetation has been restored to a certain extent, and species richness has also increased, thus improving the ecological environment in the region. The level of the ecological environmental effect status index has increased. However, the loss of population and the corresponding increase in the number of people and land has led to a decline in the level of the resource effect state index; the extensive use of resources and the improvement of environment quality coexist, making the overall village present a relatively stable evolutionary trend in terms of the ecological environmental effect.

3.  Endogenous power-driven type: Gucheng Village in the study area is a characteristic village, famous for its excellent natural geographical conditions and ancient-style buildings. The development of the "Flower Field Story" planning project in the area is based on high-quality agricultural products and rural leisure and health as supplements. Artistic landscape design and new business forms integrate and revive the countryside, creating a modern rural complex integrating ecological agriculture, leisure and health care, and rural communities, and actively promoting the development of tourism. The social effect status index is increasing, but the local villagers' construction activities lack high-quality planning and guidance, and are limited to the management level and construction experience of the township government, which leads to a negative evolution trend of the ecological environmental effect of land use.

*4.2. Analysis of the Change Trend of Ecological Environmental Effect in 2030*

Observing the prediction results of the change trend of the ecological environment effect in the study area in 2030, under the developmental orientation (DO) scenario, only two villages have improved levels of the ecological environmental effect status index, and the ecological environmental effect trend type of six of the villages is slightly negative or significantly negative. In the context of ecological orientation (EO), the number of villages with a positive evolution of the ecological environmental effect increased to three, and three villages remained stable. The ecological environmental effect trend type of Wangjiachong Village was slightly negative. Compared with the DO scenario, the EO scenario could better maintain the stability of the ecological environmental effect of land use in each village and could promote the formation of an ecological security pattern to a certain extent.

Further comparing the index value of the ecological environmental effect status of each village in 2030 with the index value of 2014 under the EO scenario, it was found that the positive evolution trend of the ecological environmental effect was not obvious: only three villages achieved a leap in the type of ecological environmental effect status (e.g., "stable" to "good" or from "degraded" to "stable"). Although the EO scenario could better maintain the stability of the ecological environmental effects than the DO scenario, the promotion of the positive evolution of the ecological environmental effect was still limited. This result shows that the strict protection of ecological land in land use management has a limited effect on improving ecological security, but the effect is limited. To achieve a higher level of ecological security, it is necessary to further coordinate land use arrangements, optimize spatial patterns, and explore more ecological security guarantees (Figure 10).

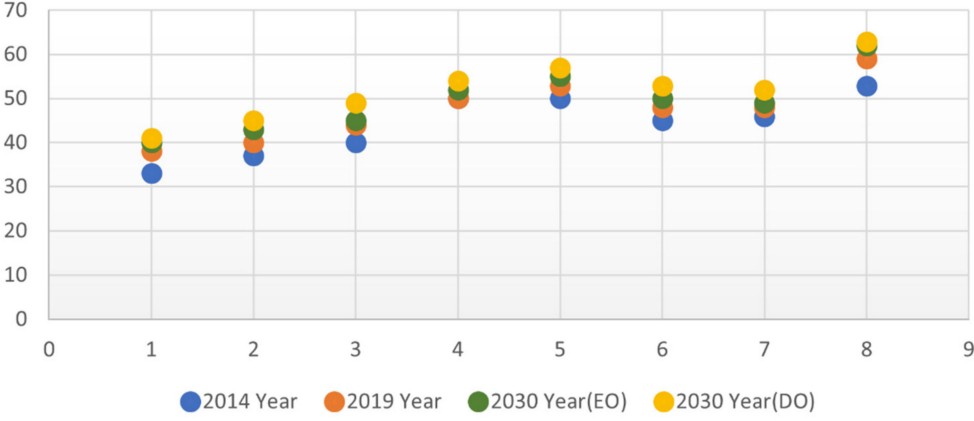

**Figure 10.** Changes in the ecological effects of land use in various townships from 2013 to 2030.

## 5. Conclusions

This paper first distinguished the concept of ecological effects and then analyzed the ecological environmental effect. We believe that the evaluation of ecological environmental effects is a comprehensive assessment of the stability of the ecosystem, focusing on the sustainability and relative safety of the ecosystem. Land is one of the key components of the ecosystem, and its changes have a profound impact on other components and ecological processes in the ecosystem. Second, assessing the ecological environmental effect of land use and predicting its changing trends are the bases for implementing ecological security safeguards from a spatial perspective. A case study of the land use of Yudian town was the starting point and we constructed the ecological environmental effect status index and the ecological environmental effect trend index, combined with land use planning, aiming at the two different scenarios of developmental orientation (DO) and ecological orientation (EO). Based on the results from 2014 to 2019, the ecological environmental effect was evaluated and predicted in 2030, and this information was used to predict the change trend of the ecological environmental effect of land use in the context of township development. Our research indicates the following results:

1.  Overall, the human activities of villages with poor physical and geographical conditions are disturbed, and they maintain a better state and level of ecological environmental effect during the development process. On the contrary, towns and towns with improved resources, environment, and economic effect endowments, due to continuous construction activities and lack of effective control, have a higher average level of environmental effect status.

2.  The villages with different development models have obvious characteristic differences in the ecological environmental effect of land use: both the village agglomeration mode type (economic effect state index) and the endogenous power leading villages (social effect state index) show negative evolution of ecological environmental effect trend, and slowly declining villages (resource effect state index and environmental effect state index) maintain relatively stable ecological environment effects during the development process; however, the resource effect state index shows a negative evolution trend due to population loss and other reasons.

3.  Traditional land use management methods, such as basic farmland protection and restricting the conversion of ecological land to construction land, have a certain effect, but made little difference. It is necessary to further coordinate land use arrangements and optimize the spatial pattern to reduce the stress of village development on the ecological environment and avoid ecological risks caused by unreasonable land use.

## 6. Discussion

There are still large regional differences in the coupling and coordination relationship among the resources, environment, society, and economy of land use in the eight villages. With the promulgation of the outline of the development plan for the Yangtze River Economic Belt, the promotion of its development requires consideration of the long-term interests of the Chinese nation, and takes the path of ecological priority and green development. Therefore, targeted promotion of the coordinated development of land use resources, environment, society, and economy effect status index between cities and towns has become an inevitable choice for future land use in Hubei Province. In this process, the differences between state indexes should form their own targeted development paths because the development mechanisms are also different (Table 2).

### 6.1. The Impact of Land Use and Cover Change on Yudian Town under the Index of Resource Effect State Index

In Table 2, the eight villages have different impacts on the resource effect status index type. Among these, the most influential village indicators are the per capita construction land area and the per capita arable land area, each accounting for five villages. The natural resource endowments of different villages are different, the regional development goals are diversified, the land use is multi-layered, and the pattern of land use function change shows obvious spatial differentiation characteristics. According to the questionnaire survey, the average household area of construction land accounts for 1 hm$^2$, and the area of cultivated land per household accounts for 0.2 hm$^2$. Due to the development of the economy and the continuous increase in population, especially the acceleration of industrialization and urbanization, the demand for land resources is increasing each year, leading to the occupation of a large amount of arable land by construction and a shortage of arable land reserve resources. Furthermore, unreasonable land development and utilization have reduced the stability of the ecosystem in the region. Taking Jinpan Village and Shuanglou Village as examples, it is necessary to improve the effect of land use resources, promote green and clean production and circular economy development, focus on strengthening the sustainable development of resource-based industries, vigorously support the development of the tertiary industry, and promote the structure of rural industries. Upgrading and optimizing land use to strengthen investment in the ecological environment will enhance the ecological effect of land use.

**Table 2.** The impact of eight villages on the ecological environment state index.

| Each Index | Index | Wangjiachong Village | Bai Que Village | Yingzizhai Village | Jinpan Village | Lufan Village | Gu Cheng Village | Baozi Ling Village | Shuanglou Village |
|---|---|---|---|---|---|---|---|---|---|
| Resource Effect State Index | Intensity of national land development | √ | | | √ | | | √ | |
| | Per capita construction land area | | | √ | √ | | √ | √ | √ |
| | Per capita arable land area | √ | √ | | √ | √ | | | √ |
| | Per capita forest area | | | √ | | √ | | √ | |
| Environmental Effect State Index | Water network density | | √ | √ | | | √ | | |
| | Vegetation coverage | √ | | | √ | √ | √ | √ | √ |
| | Land degradation | | √ | | √ | | √ | | |
| | Biodiversity | √ | √ | √ | | √ | | √ | |
| Social Effect State Index | Population density | | | √ | | | | √ | √ |
| | Population natural growth rate | | √ | √ | √ | √ | | | |
| | Engel coefficient | √ | √ | | √ | | √ | √ | √ |
| Economic Effect State Index | Land for transportation | | √ | | | √ | √ | | |
| | Per capita public budget input | √ | √ | √ | √ | √ | √ | √ | √ |
| | Per capita GDP value | √ | √ | | | | √ | √ | |

*6.2. The Impact of Land Use and Cover Change on Yudian Town under the Environmental Effect State Index*

The main factors affecting the environmental effect status index are vegetation coverage and biodiversity, of which vegetation coverage accounts for six villages and biodiversity accounts for four villages. According to field investigations, Lufan Village, Gucheng Village, and Baoziling Village make full use of the excellent ecological environment advantages, strictly protect the natural ecological pattern of the landform, build a green ecological network, and promote the interactive development of agricultural production and characteristic tourism. The land use functions of Baique Village and Jinpan Village both show different levels of weakening, mainly in the aspect of land degradation. Due to the shortage of land resources in the villages, the tightening of resource constraints, and the environmental carrying capacity approaching the upper limit, Baique Village and Jinpan Village are affected by unreasonable development and utilization, and the functions of ecological environment purification are reduced.

*6.3. The Impact of Land Use and Cover Change on Yudian Town under the Social Effect State Index*

The main indicator that affects the social effect state index is the Engel coefficient, which accounts for six villages. The average in the six villages is 40%. According to the results, the population density and the natural population growth rate show a negative trend, which is mainly reflected in the transformation of the township population to urbanization. The economic and social development within the town and the agglomeration of population and industries are the main reasons affecting population mobility. Under the new situation of urbanization and ecological civilization construction, land use functions need to go beyond ensuring basic human survival and living needs, and pay more attention to meeting the needs of the sustainable and coordinated development of humans and nature. The transformation of the coordinated development of functions is required. Gucheng Village is an example; while the village vigorously develops characteristic tourism, it must also enhance the ability of the region to transform production and living elements, which is conducive to the improvement of the social effect of land use in the region.

*6.4. The Impact of Land Use and Cover Change on Yudian Town under the Economic Effect State Index*

In Table 2, the eight villages all have an impact on the average annual net income of farmers in the economic effect status index. Taking Wangjiachong Village, Baique Village, Gucheng Village and Baoziling Village as examples, the average GDP per capita of the four villages is RMB 10,000 or more, with outstanding industrialization development and large-scale tourism, vigorous development of the export-oriented economy and modern service industry, and promotion of the continuous agglomeration of population and non-agricultural industries in the transportation areas. Thus, the development of the economic value of local villages is improving. Compared with the disadvantages of other villages, it is necessary to realize the improvement of the economic effect of land use, build a green industrial development system, and promote the healthy development of the economy. On this basis, the investment capacity of infrastructure and other elements can be improved and the economic effect of land use enhanced.

The evaluation of the ecological environment effect of land use is carried out in different regions. For each specific village, the development conditions of the village are strongly regional due to the different natural resource endowments and ecosystems in the different regions. This result also determines the localized nature of ecological environmental effect evaluation research. Due to the weak research on the ecological environmental effect evaluation in the low hilly area of northern Hubei, taking a study area as an example does not represent the ecological environmental effect evaluation system of the entire region. In addition, the village statistics foundation is weak, the data acquisition is incomplete, and the available dataset is relatively small. The data support and index refinement of the research have caused certain restrictions. When it is extended to other



regions for research, the evaluation index needs to be further expanded and deepened. Therefore, in future research it is necessary to further study and construct an evaluation system of the ecological environmental effect of land use focusing on the evaluation of the stability of the ecosystem, and determine the change trend of the ecological environmental effect mainly according to the change in the state index.

**Author Contributions:** Conceptualization, methodology, software, validation, formal analysis, data curation, writing—original draft preparation, visualization, writing–editing: W.R.; and Y.S. project administration, funding, review: W.R.; and X.Z. All authors have read and agreed to the published version of the manuscript.

**Funding:** This research is supported by the National Natural Science Foundation of China (grant number 42071455) and the special scientific research project of Hubei Provincial Land Consolidation Bureau in 2019.

**Conflicts of Interest:** The authors declare no conflict of interest.

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
