# Peer review of "Evaluation of Ecological Environment Effect of Villages Land Use and Cover Change: A Case Study of Some Villages in Yudian Town, Guangshui City, Hubei Province"

_land, doi:10.3390/land10030251_

Round 1
Reviewer 1 Report
This is an important study on an important subject, how to make more ecologically rational planning decisions in peri urban areas in China.
I my view this paper is a good start and that the authors might consider applying for additional grant funds to develop the indices better, and to look at changes over a longer time frame with a view of making the methodology more applicable over other regions of china

Author Response
Dear Editors and Reviewers:
The paper has been revised as required, please see the PDF.

Reviewer 2 Report
Comments and Suggestions are attached (.docx).

Author Response
Dear Editors and Renviewers:
The paper has been revised as required, please see the PDF
